# Bayesian Robust Graph Contrastive Learning

## Abstract

Graph Neural Networks (GNNs) have been widely used to learn node representations and with outstanding performance on various tasks such as node classification. However, noise, which inevitably exists in real-world graph data, would considerably degrade the performance of GNNs revealed by recent studies. In this work, we propose a novel and robust method, Bayesian Robust Graph Contrastive Learning (BRGCL), which trains a GNN encoder to learn robust node representations. The BRGCL encoder is a completely unsupervised encoder. Two steps are iteratively executed at each epoch of training the BRGCL encoder: (1) estimating the confident nodes and computing the robust cluster prototypes of the node representations through a novel Bayesian nonparametric method; (2) prototypical contrastive learning between the node representations and the robust cluster prototypes. Experiments on public benchmarks demonstrate the superior performance of BRGCL and the robustness of the learned node representations. The code of BRGCL is available at `https://anonymous.4open.science/r/BRGCL-code-2FD9/`.

## 1    Introduction

Graph Neural Networks (GNNs) have become popular tools for node representation learning in recent years (Kipf & Welling, 2017; Bruna et al., 2014; Hamilton et al., 2017; Xu et al., 2019). Most prevailing GNNs (Kipf & Welling, 2017; Zhu & Koniusz, 2021) leverage the graph structure and obtain the representation of nodes in a graph by utilizing the features of their connected nodes. Benefiting from such propagation mechanism, node representations obtained by GNN encoders have demonstrated superior performance on various downstream tasks such as semi-supervised node classification and node clustering.

Although GNNs have achieved great success in node representation learning, current GNN approaches do not consider the noise in the input graph. However, noise inherently exists in the graph data for many real-world applications. Such noise may be present in node attributes or node labels, which forms two types of noise, attribute noise and label noise. Recent works, such as (Patrini et al., 2017), have evidenced that noisy inputs hurt the generalization capability of neural networks. Moreover, noise in a subset of the graph data can easily propagate through the graph topology to corrupt the remaining nodes in the graph data. Nodes that are corrupted by noise or falsely labeled would adversely affect the representation learning of themselves and their neighbors.

While manual data cleaning and labeling could be remedies to the consequence of noise, they are expensive processes and difficult to scale, thus not able to handle the almost infinite amount of noisy data online. Therefore, it is crucial to design a robust GNN encoder that could make use of noisy training data while circumventing the adverse effect of noise. In this paper, we propose a novel and robust method termed Bayesian Robust Graph Contrastive Learning (BRGCL) to improve the robustness of node representations for GNNs. Our key observation is that there exist a subset of nodes which are confident in their class/cluster labels. Usually, such confident nodes are far away from the class/cluster boundaries, so these confident nodes are trustworthy, and noise in these nodes would not degrade the value of these nodes in training a GNN encoder. To infer such confident nodes, we propose a novel algorithm named Bayesian nonparametric Estimation of Confidence (BEC). Since the BRGCL encoder is completely unsupervised, it first infers pseudo labels of all the nodes with a Bayesian nonparametric method only based on the input node attributes, without knowing the ground truth labels or the ground truth class number in the training data. Then, BEC

is used to estimate the confident nodes based on the pseudo labels and the graph structure. The robust prototypes, as the cluster centers of the confident nodes, are computed and used to train the BRGCL encoder with a loss function for prototypical contrastive learning. The confident nodes are updated during each epoch of the training of the BRGCL encoder, so the robust prototypes are also updated accordingly.

## 1.1 Contributions

Our contributions are as follows.

First, we propose Bayesian Robust Graph Contrastive Learning (BRGCL), where a *fully unsupervised* encoder is trained on noisy graph data. The fully unsupervised BRGCL encoder is trained only on the input node attributes without ground truth labels and even the ground truth class number in the training data. BRGCL leverages confident nodes, which are estimated by a novel algorithm termed Bayesian nonparametric Estimation of Confidence (BEC), to harvest noisy graph data without being compromised by the noise. Experimental results on popular graph datasets evidence the advantage of BRGCL over competing GNN methods for node classification and node clustering on noisy graph data. The significance of the improvement of BRGCL is evidenced by $p$-values of t-test.

Second, our study reveals the importance of confident nodes in training GNN encoders on noisy graph data, which opens the door for future research in this direction. The visualization results in Section 5.6 show that the confident nodes estimated by BEC are usually far away from the class/cluster boundaries, and so are the robust prototypes. As a result, the BRGCL encoder trained with such robust prototypes is not vulnerable to noise, and it even outperforms GNNs trained with ground truth labels.

## 2 Related Works

### 2.1 Graph Neural Networks

Graph neural networks (GNNs) have become popular tools for node representation learning. They have shown superior performance in various graph learning tasks, such as node classification, node clustering, and graph classification. Given the difference in the convolution domain, current GNNs fall into two classes. The first class features spectral convolution (Bruna et al., 2014; Kipf & Welling, 2017), and the second class (Hamilton et al., 2017; Velickovic et al., 2018; Xu et al., 2019) generates node representations by sampling and propagating features from their neighborhood. GNNs such as ChebNet (Bruna et al., 2014) perform convolution on the graph Fourier transforms termed spectral convolution. Graph Convolutional Network (GCN) (Kipf & Welling, 2017) further simplifies the spectral convolution (Bruna et al., 2014) by its first-order approximation. GNNs such as GraphSAGE (Hamilton et al., 2017) propose to learn a function that generates node representations by sampling and propagating features from a node's connected neighborhood to itself. Various designs of the propagation function have been proposed. For instance, Graph Attention Network (GAT) (Velickovic et al., 2018) proposes to learn masked self-attention layers that enable nodes to attend over their neighborhoods' features. Different from GNNs based on spectral convolution, such methods could be trained on mini-batches (Hamilton et al., 2017; Xu et al., 2019), so they are more scalable to large graphs.

However, as pointed out by (Dai et al., 2021), the performance of GNNs can be easily degraded by noisy training data (NT et al., 2019). Moreover, the adverse effects of noise in a subset of nodes can be exaggerated by being propagated to the remaining nodes through the graph structure, exacerbating the negative impact of noise.

To learn node representation without node labels, contrastive learning has recently been applied to the training of GNNs. Most proposed graph contrastive learning methods create multiple views of the unlabeled input graph and maximize agreement between the node representations of these views. Deep Graph Infomax (DGI)(Velickovic et al., 2019) contrasts high-level graph representations by maximizing their mutual information. Following that, InfoGraph(Sun et al., 2020) obtains representations for entire graphs by contrasting graph-level representations with substructure-level representations. Other methods, such as MVGRL (Hassani & Ahmadi, 2020), GCC (Qiu et al., 2020), and GRACE (Zhu et al., 2020), contrast augmented views of the graph generated by various techniques, such as graph diffusion, subgraph sampling,

edge dropping, and feature masking. MERIT (Jin et al., 2021) performs contrastive learning across augmented views generated by siamese networks. SUGRL (Mo et al., 2022) proposes a multiplet loss to learn the complementary information between the structural information and neighbor information via contrastive learning. Unlike previous methods, we propose using contrastive learning to train GNN encoders that are robust to noise existing in the labels and attributes of nodes.

## 2.2 Existing Methods Handing Noisy Data

Previous works (Zhang et al., 2021) have shown that deep neural networks usually generalize badly when trained on input with noise. Existing literature on robust learning with noisy inputs mostly focuses on image or text domain. Such robust learning methods fall into two categories. The first category (Patrini et al., 2017; Goldberger & Ben-Reuven, 2017) mitigates the effects of noisy inputs by correcting the computation of loss function, known as loss corruption. The second category aims to select clean samples from noisy inputs for the training (Malach & Shalev-Shwartz, 2017; Jiang et al., 2018; Yu et al., 2019; Li et al., 2020; Han et al., 2018), known as sample selection. For example, (Goldberger & Ben-Reuven, 2017) corrects the predicted probabilities with a corruption matrix computed on a clean set of inputs. On the other hand, recent sample selection methods usually select a subset of training data to perform robust learning. Among the existing loss correction and sample selection methods, Co-teaching (Han et al., 2018) is promising, which trains two deep neural networks and performs sample selection in a training batch by comparing predictions from the two networks. However, such sample selection strategy does not generalize well in the graph domain (Dai et al., 2021) due to the extraordinarily small size of labeled nodes. More details are to be introduced in Section 4.2. Self-Training (Li et al., 2018) finds nodes with the most confident pseudo labels, and it augments the labeled training data by incorporating confident nodes with their pseudo labels into the existing training data. In addition to the above two categories of robust learning methods, recent studies (Kang et al., 2020; Zhong et al., 2021; Wang et al., 2021) show that decoupling the feature representation learning and the training of the classifier can also improve the robustness of the learned feature representation.

NRGNN(Dai et al., 2021) proposes a novel graph neural network model for semi-supervised node classification on sparsely and noisily labeled graphs. It introduces a graph edge predictor to predict missing links for connecting unlabeled nodes with labeled nodes, and a pseudo label miner to expand the label set. RTGNN (Qian et al., 2023) also aims to train a robust GNN classifier with scarce and noisy node labels. It first classifies labeled nodes into clean and noisy ones and adopts reinforcement supervision to correct noisy labels. It also generates pseudo labels to provide extra training signals. During the training of the node classifier, RTGNN also introduces a consistency regularization term to prevent overfitting to noise. To improve the robustness of the node classifier on the dynamic graph, GraphSS (Zhuang & Hasan, 2022) proposes to generalize noisy supervision as a kind of self-supervised learning method, which regards the noisy labels, including both manual-annotated labels and auto-generated labels, as one kind of self-information for each node. They show that the robustness of the node classifier can be improved by utilizing such self-information in self-supervised learning. In addition, some recent works (Dai et al., 2022) seek to train robust node classifiers with noisy graphs. To learn robust GNNs on noisy graphs with limited labeled nodes, RS-GNN (Dai et al., 2022) adopts the edges in the noisy graph as supervision to obtain a denoised and densified graph to facilitate the message passing for predictions of unlabeled nodes.

## 2.3 Prototypical Learning

By introducing the means of embedded samples within data clusters as prototypes, prototypical learning has been recently applied to the learning of neural network methods. Based on the usage of the prototypes in the learning regime, current prototypical learning methods can be classified into two categories. Methods in the first category aim to learn prototype-based classifiers with limited labeled data. For instance, to handle the issue of limited supervision for new categories in few-shot learning, prototypical network (Snell et al., 2017) learns the prototypes of new classes in the embedding space, where the classification is done by calculating the distance between the test image and prototypes of each class. Such a prototype-based classification framework is further proven to be more robust for zero-shot learning (Allen et al., 2019; Xu et al., 2020) and out-of-distribution learning (Arik & Pfister, 2020) as well. The other category of prototypical learning methods aims to encode semantic structures into the embedding space. PCL (Li et al., 2021a)

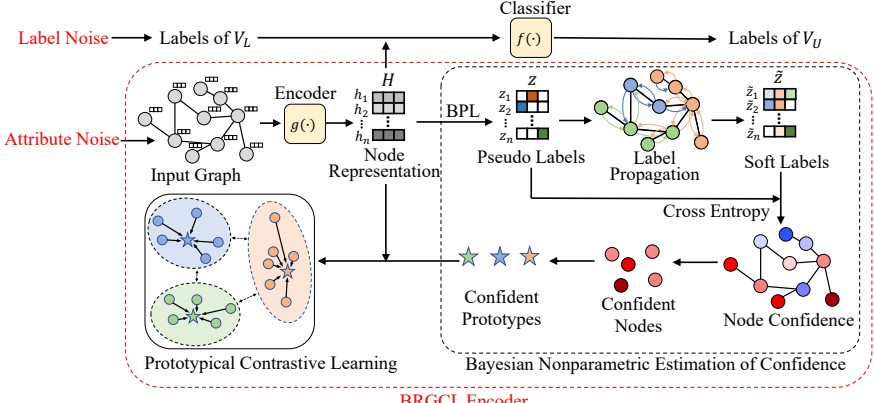

Figure 1: Illustration of the BRGCL encoder. BPL stands for the Bayesian nonparametric Prototype Learning to be introduced in Section 4.2, a Bayesian nonparametric algorithm for estimating the pseudo labels of nodes. In the illustration of confident nodes, nodes which are more confident in their pseudo labels are marked in more red, and less confident nodes are marked in more blue.

first introduces prototypes as latent variables in contrastive learning to encode the semantic structure of data explicitly. An EM algorithm is designed to perform representation learning and clustering iteratively. Next, HCSC (Guo et al., 2022) argues that semantic structures in an image dataset are hierarchical in nature, as a result, they propose to learn a set of hierarchical prototypes in contrastive learning instead of representing each class with a single prototype. Following that, GraphLoG (Xu et al., 2021) introduces prototype learning in self-supervised graph-level representation learning to learn the global semantics of graph structure. Similar to HCSC, GraphLoG also learns a set of hierarchical prototypes in the representation space for graph classification.

In contrast with existing prototype learning methods, our work proposes to learn clean and robust prototypes from the confident nodes, and the prototypical loss encourages the learned node features to be aligned with the robust prototypes.

## 3 Problem Setup

### 3.1 Notations

An attributed graph consisting of $N$ nodes is formally represented by $\mathcal{G} = (\mathcal{V}, \mathcal{E}, \mathbf{X})$, where $\mathcal{V} = \{v_1, v_2, \ldots, v_N\}$ and $\mathcal{E} \subseteq \mathcal{V} \times \mathcal{V}$ denote the set of nodes and edges respectively. $\mathbf{X} \in \mathbb{R}^{N \times D}$ are the node attributes, and the attributes of each node is in $\mathbb{R}^d$. Let $\mathbf{A} \in \{0,1\}^{N \times N}$ be the adjacency matrix of graph $\mathcal{G}$, with $\mathbf{A}_{ij} = 1$ if and only if $(v_i, v_j) \in \mathcal{E}$. $\tilde{\mathbf{A}} = \mathbf{A} + \mathbf{I}$ denotes the adjacency matrix for a graph with self-loops added. $\tilde{\mathbf{D}}$ denotes the diagonal degree matrix of $\tilde{\mathbf{A}}$. Let $\mathcal{V}_{\mathcal{L}}$ and $\mathcal{V}_{\mathcal{U}}$ denote the set of labeled nodes and unlabeled nodes respectively. Throughout this paper, we use $\|\cdot\|_2$ to denote the Euclidean norm of a vector and $[n]$ to denote all the natural numbers between 1 and $n$ inclusively.

### 3.2 Graph Convolution Network (GCN)

To learn the node representations from the attributes $\mathbf{X} \in \mathbb{R}^{N \times d}$ and the graph structure $\mathbf{A}$, one simple yet effective neural network model is Graph Convolution Network (GCN). GCN is originally proposed for semi-supervised node classification, which comprises two graph convolution layers. In our work, we use GCN as the encoder to obtain node representation $\mathbf{H} \in \mathbb{R}^{N \times M}$, where the $i$-th row of $\mathbf{H}$ is the node representation of $v_i$. The GCN encoder has two layers which can be formulated as

$$\mathbf{H} = \sigma(\hat{\mathbf{A}}\sigma(\hat{\mathbf{A}}\mathbf{X}\mathbf{W}^{(0)})\mathbf{W}^{(1)}), \tag{1}$$

where $\hat{\mathbf{A}} = \tilde{\mathbf{D}}^{-1/2}\tilde{\mathbf{A}}\tilde{\mathbf{D}}^{-1/2}$ is the normalized graph Laplacian, $\mathbf{W}^{(0)}, \mathbf{W}^{(1)} \in \mathbb{R}^{d \times m}$ are the weight matrices with hidden dimension $m$, and $\sigma$ is the activation function of ReLU.

### 3.3 Problem Description

Noise usually exists in the input node attributes or labels of real-world graphs, which degrades the quality of the node representations obtained by common GCL encoders and adversely affects the performance of the classifier trained on such representations. We aim to obtain node representations robust to noise in two cases, where noise is present in either the labels of $\mathcal{V}_\mathcal{L}$ or in the input node attributes $\mathbf{X}$. That is, we consider either noisy label or noisy input node attributes.

The goal of BRGCL is to learn a node representation $\mathbf{H} = g(\mathbf{X}, \mathbf{A})$, such that the node representations $\{\mathbf{h}_i\}_{i=1}^N$ are robust to noise in the above two cases, where $g(\cdot)$ is the BRGCL encoder in (1). To evaluate the performance of the robust node representations by BRGCL, the node representations $\{\mathbf{h}_i\}_{i=1}^N$ are used for the following two tasks.

(1) Semi-supervised node classification, where a classifier is trained on $\mathcal{V}_\mathcal{L}$ and the class labels of $\mathcal{V}_\mathcal{L}$, and then the classifier predicts the labels of the unlabeled nodes $\mathcal{V}_\mathcal{U}$.

(2) Node clustering, where K-means clustering is performed on the node representations $\{\mathbf{h}_i\}_{i=1}^N$ to obtain node clusters.

## 4 Bayesian Robust Graph Contrastive Learning

We propose Bayesian Robust Graph Contrastive Learning (BRGCL) in this section to improve the robustness of node representations. First, we review the preliminaries of graph contrastive learning. Next, we propose a new Bayesian nonparametric Estimation of Confidence (BEC) algorithm to estimate robust nodes and prototypes. Then we show details of node classification and node clustering. At last, we propose a decoupled training pipeline of BRGCL for semi-supervised node classification. Figure 1 illustrates the overall framework of BRGCL.

### 4.1 Preliminary of Graph Contrastive Learning

The general node representation learning aims to train an encoder $g(\cdot)$, which is a two-layer Graph Convolution Neural Network (GCN) (Kipf & Welling, 2017), to generate discriminative node representations. In our work, we adopt contrastive learning to train the BRGCL encoder $g(\cdot)$. To perform contrastive learning, two different views, denoted by $G^1 = (\mathbf{X}^1, \mathbf{A}^1)$ and $G^2 = (\mathbf{X}^2, \mathbf{A}^2)$, are generated by node dropping, edge perturbation, and attribute masking. The representations of the two generated views are denoted as $\mathbf{H}^1 = g(\mathbf{X}^1, \mathbf{A}^1)$ and $\mathbf{H}^2 = g(\mathbf{X}^2, \mathbf{A}^2)$, with $\mathbf{h}_i^1$ and $\mathbf{h}_i^2$ being the $i$-th row of $\mathbf{H}^1$ and $\mathbf{H}^2$ respectively. It is preferred that the mutual information between $\mathbf{H}^1$ and $\mathbf{H}^2$ is maximized. For computational efficiency its lower bound is usually used as the objective for contrastive learning. We use InfoNCE (Li et al., 2021a) as our node-wise contrastive loss, that is,

$$\mathcal{L}_{\text{node}} = \sum_{i=1}^N -\log \frac{s(\mathbf{h}_i^1, \mathbf{h}_i^2)}{s(\mathbf{h}_i^1, \mathbf{h}_i^2) + \sum_{j=1}^N s(\mathbf{h}_i^1, \mathbf{h}_j^2)}, \tag{2}$$

where $s(\mathbf{h}_i^1, \mathbf{h}_i^2) = \frac{|\langle \mathbf{h}_i^1, \mathbf{h}_i^2 \rangle|}{\|\mathbf{h}_i^1\|_2 \|\mathbf{h}_i^2\|_2}$ is the cosine similarity between two node representations $\mathbf{h}_i^1$ and $\mathbf{h}_i^2$.

In addition to the node-wise contrastive learning, we also adopt prototypical contrastive learning (Li et al., 2021a) to capture semantic information in the node representations, which can be interpreted as maximizing the mutual information between node representation and a set of estimated cluster prototypes $\{\mathbf{c}_1, ..., \mathbf{c}_K\}$. Here $K$ is the number of cluster prototypes. The loss function for prototypical contrastive learning is

$$\mathcal{L}_{\text{proto}} = -\frac{1}{N} \sum_{i=1}^N \log \frac{\exp(\mathbf{h}_i \cdot \mathbf{c}_k / \tau)}{\sum_{k=1}^K \exp(\mathbf{h}_i \cdot \mathbf{c}_k / \tau)} \tag{3}$$

BRGCL aims to improve the robustness of node representations by prototypical contrastive learning. Our key observation is that there exists a subset of nodes that are confident about their class/cluster labels because they are far away from the class/cluster boundaries. We propose an effective method to infer such confident nodes. Because the BRGCL encoder is completely unsupervised, it does not have access to the ground truth label or ground truth class/cluster number. Therefore, our algorithm for selection of confident nodes is based on a Bayesian non-parameter styled inference, and the algorithm is termed Bayesian nonparametric Estimation of Confidence (BEC) to be introduced next.

### 4.2 Bayesian nonparametric Estimation of Confidence (BEC)

The key idea of Bayesian nonparametric Estimation of Confidence (BEC) is to estimate robust nodes by the confidence of nodes in their labels. Intuitively, nodes more confident in their labels are less likely to be adversely affected by noise. Because BRGCL is unsupervised, pseudo labels are used as the labels for such estimation.

We propose Bayesian nonparametric Prototype Learning (BPL) to infer the pseudo labels of nodes. BPL, as a Bayesian nonparametric algorithm, infers the pseudo labels by the Dirichlet Process Mixture Model (DPMM) under the assumption that the distribution of the node representations is a potentially infinite mixture of Gaussians. We assume each prototype corresponds to a Gaussian component. Let $p(\boldsymbol{h}) = \sum_{c=1}^{K} \pi_k \mathcal{N}(\boldsymbol{h} \mid \mathbf{c}_k, \Sigma_k)$ be the density of the distribution for the node representations, where $K$ is the number of Gaussian components, $\{\pi_k\}_{k=1}^{K}$ are the mixing coefficients, and $\{\mathbf{c}_c\}_{k=1}^{K}$ and $\{\Sigma_c\}_{k=1}^{K}$ are the means and covariances respectively. The data generation process can be described as a generative model in which a component is selected based on the probability $\pi_k$, followed by the generation of an observation from the Gaussian associated with the selected component. Let $\text{Dir}(K, \boldsymbol{\pi}_0)$ be a Dirichlet prior on the mixing coefficients for some $\boldsymbol{\pi}_0$. Assuming that the covariances of all the Gaussian components are constrained to $\sigma I$, and the means are sampled from some prior distribution $G_0$, the DPMM can be described by the following model in (Kulis & Jordan, 2012):

$$\mathbf{c}_1, ..., \mathbf{c}_k \sim G_0, \quad \boldsymbol{\pi} \sim \text{Dir}(K, \boldsymbol{\pi}_0), \quad z_1, ..., z_n \sim \text{Discrete}(\boldsymbol{\pi}), \quad \boldsymbol{x}_1, ..., \boldsymbol{x}_n \sim \mathcal{N}(\mathbf{c}_{z_i}, \sigma I), \tag{4}$$

with $K \to \infty$. Following (Kulis & Jordan, 2012), Gibbs sampling can be used to iteratively sample the pseudo labels for each node representation given the means of all the Gaussian components, and sample the means of the Gaussian components given the pseudo labels of all the node representations. Such a process is almost equivalent to K-means when $\sigma$, the variance of the Gaussians, goes to 0. We consider the case that $\sigma \to 0$ in BPL, and the almost zero variance eliminates the need to estimate the variance $\sigma$, making the inference efficient.

Let $\tilde{K}$ denote the number of inferred prototypes at the current iteration of the Gibbs sampling process, the pseudo label $z_i$ is then calculated by

$$z_i = \arg\min_{k} \{d_{ik}\}, \, i \in [N],$$
$$d_{ik} = \begin{cases} \|\mathbf{h}_i - \mathbf{c}_k\|_2^2 & k \in [\tilde{K}], \\ \xi & k = \tilde{K} + 1, \end{cases} \tag{5}$$

where the Euclidean distance $\{d_{ik}\}$ is used to determine the pseudo label of the node representation $\mathbf{h}_i$. $\xi$ is the margin to initialize a new prototype. In practice, we choose the value of $\xi$ by performing cross-validation on each dataset.

After obtaining the pseudo labels of all the nodes by BPL with $K$ being the inferred number of prototypes, we estimate the confidence of the nodes based on their pseudo labels and the graph structure. We first select the nodes confident in their labels, referred to as the confident nodes, by considering the label information from the neighborhood of each node specified by the adjacency matrix. Let $\boldsymbol{z}_i$ denote the one-hot pseudo label of node $v_i$ estimated by BPL. Label propagation (Zhang & Chen, 2018) is then applied based on the adjacency matrix to get a soft pseudo label for each node. Let $\mathbf{Z} \in \mathbb{R}^{N \times K}$ be the matrix of pseudo labels with $\mathbf{z}_i$ being the $i$-th row of $\mathbf{Z}$. The label propagation runs the following update for $T$ steps,

$$\mathbf{Z}^{(t)} = (1 - \alpha)\tilde{\mathbf{A}}\mathbf{Z}^{(t-1)} + \alpha\mathbf{Z} \quad t = 1, ..., T - 1, \tag{6}$$

where $\mathbf{Z}^{(0)} = \mathbf{Z}$, $T$ is the number of propagation steps and $\alpha$ is the teleport probability, which are set to the suggested values in (Zhang & Chen, 2018). Let $\tilde{\mathbf{Z}} = \mathbf{Z}^{(T)}$ be the soft labels obtained by the label propagation with $\tilde{z_i}$ being the $i$-th row of $\tilde{\mathbf{Z}}$. Following (Han et al., 2018), we use the cross-entropy between $z_i$ and $\tilde{z_i}$, denoted by $\phi(z_i, \tilde{z_i})$, to identify the confident nodes. Intuitively, smaller cross-entropy $\phi(z_i, \tilde{z_i})$ of a node $v_i$ indicates that the pseudo label $z_i$ is more consistent with the pseudo label suggested by the neighbors of node $v_i$, so that node $v_i$ is more confident about its pseudo label $z_i$. As a result, we can define the confidence score of a node $v_i$ on its pseudo label $z_i$ by $\alpha_i = 1 - \phi(z_i, \tilde{z_i})$. We then obtain the set of confident nodes assigned to the $k$-th cluster as

$$\mathcal{T}_k = \{ h_i \mid \alpha_i = 1 - \phi(z_i, \tilde{z_i}) > 1 - \gamma, z_i = k, i \in [N] \}, \tag{7}$$

where $\gamma$ is a threshold for the selection of confident nodes. (7) indicates that the nodes with confidence scores greater than $1 - \gamma$ are selected as the confidence nodes for each cluster. Figure 2 illustrates the normalized confidence scores of all the nodes for different levels of noise present in the input node attributes, and the normalized confidence score is defined in Section 5.6 which scales all the confidence scores into $[0, 1]$. The confident nodes with larger normalized confidence scores, which are marked in more red, are far away from cluster boundaries, so that noise on these nodes is more unlikely to affect their classification/clustering labels. These confident nodes are the robust nodes leveraged by BRGCL to fight against noise.

The threshold $\gamma$ is dynamically set by

$$\gamma = 1 - \gamma_0 \min \left\{ 1, \frac{t}{t_{\max}} \right\}, \tag{8}$$

where $t$ is the current epoch number and $t_{\max}$ is a preset number of training epochs. $\gamma_0$ is an annealing factor, which is decided by cross-validation for each dataset in practice. Previous methods such as (Li et al., 2021a) estimate each prototype as the mean of node representations assigned to that prototype. We propose to estimate each prototype only using the confident nodes assigned to that prototype for enhanced robustness. To this end, after acquiring the confident nodes $\{\mathcal{T}_k\}_{k=1}^K$, the robust prototypes are updated by $c_k = \frac{1}{|\mathcal{T}_k|} \sum_{h_i \in \mathcal{T}_k} h_i$ for each $k \in [K]$. With the updated robust prototypes $\{c_k\}_{k=1}^K$ in the prototypical contrastive learning loss $\mathcal{L}_{\mathrm{proto}}$ in (3), we train the encoder $g(\cdot)$ with the following overall loss function,

$$\mathcal{L}_{\mathrm{rep}} = \mathcal{L}_{\mathrm{node}} + \mathcal{L}_{\mathrm{proto}}. \tag{9}$$

In Co-teaching (Han et al., 2018), a threshold similar to (8) is used to select a ratio of data for training. However, due to the limited size of training data in graph domain, training with only a subset of nodes usually leads to degraded performance. For example, with 5% of nodes labeled on the Cora dataset, only 1% of all the nodes will be used for training if the threshold is set to 20% by Co-teaching. In contrast, BEC selects confident nodes by a dynamic threshold on the confidence scores of the nodes. The selected confident nodes are only used to obtain the robust prototypes, and BRGCL is trained with such robust prototypes to obtain robust representations for all the nodes of the graph. It is also worthwhile to mention that training BRGCL with the loss function $\mathcal{L}_{\mathrm{rep}}$ does not require any information about the ground truth labels. The training algorithm for the BRGCL encoder is described in Algorithm 1. It is noted that the confident nodes and the robust prototypes are estimated at each epoch by BEC.

### 4.3 Decoupled Training

The typical pipeline for semi-supervised node classification is to jointly train the classifier and the encoder. However, the noise in the training data would degrade the performance of the classifier. To alleviate this issue, we decouple the representation learning for the nodes from the classification of nodes to mitigate the effect of noise, which consists of two steps. In the first step, the BRGCL encoder $g(\cdot)$ is trained by Algorithm 1 to obtain the node representation $\mathbf{H}$. We then build a node classifier $f(\cdot)$ as a two-layer MLP followed by a softmax function. In the second step, the classifier $f(\cdot)$ is trained by minimizing the cross-entropy loss on the labeled nodes, $\mathcal{L}_{\mathrm{cls}} = \frac{1}{|\mathcal{V}_{\mathcal{L}}|} \sum_{v_i \in \mathcal{V}_{\mathcal{L}}} H(\tilde{y}_i, f(\mathbf{h}_i))$, where $H$ is the cross-entropy function and $\tilde{y}_i$ is the ground truth class label for node $v_i$. In Section 5.5, we show the advantage of such decoupled learning pipeline over the conventional joint training of the encoder and the classifier.

---
**Algorithm 1** Training algorithm of BRGCL encoder

---
**Input:** The input attribute matrix $\mathbf{X}$, adjacency matrix $\mathbf{A}$, learning rate $\eta$, and the training epochs $t_{\max}$.
**Output:** The parameter of BRGCL encoder $g$.
 1: Initialize the parameter $\mathbf{W} = (\mathbf{W}^{(0)}, \mathbf{W}^{(1)})$ of BRGCL encoder $g$
 2: **for** $t \leftarrow 1$ to $t_{\max}$ **do**
 3:    Calculate node representations by $\mathbf{H} = g(\mathbf{X}, \mathbf{A})$
 4:    Generate augmented views $G^1 = (\mathbf{X}^1, \mathbf{A}^1)$ and $G^2 = (\mathbf{X}^2, \mathbf{A}^2)$
 5:    Calculate node representations of augmented views by $\mathbf{H}^1 = g(\mathbf{X}^1, \mathbf{A}^1)$ and $\mathbf{H}^2 = g(\mathbf{X}^2, \mathbf{A}^2)$
 6:    Obtain the pseudo labels $\mathbf{Z}$ of all the nodes and the number of inferred prototypes $K$ by BPL
 7:    Obtain the soft labels $\tilde{\mathbf{Z}}$ of all the nodes by label propagation described in (6)
 8:    Update the confidence threshold $\gamma$ by (8)
 9:    Estimate the confident nodes $\{\mathcal{T}_k\}_{k=1}^K$ by (7)
10:    Update the robust prototypes by $\boldsymbol{c}_k = \frac{1}{|\mathcal{T}_k|} \sum_{\boldsymbol{h}_i \in \mathcal{T}_k} \boldsymbol{h}_i$ for all $k \in [K]$
11:    Update the parameter $\mathbf{W}$ of the BRGCL encoder $g$ by $\mathbf{W} \leftarrow \mathbf{W} - \eta \nabla_{\mathbf{W}} \mathcal{L}_{\text{rep}}$ with $\mathcal{L}_{\text{rep}}$ described in (9)
12: **end for**
13: **return** The BRGCL encoder $g$

---

## 5 Experiments

In this section, we evaluate the performance of BRGCL on five public benchmarks. For semi-supervised node classification, the performance of BRGCL is evaluated with noisy labels or noisy input node attributes. For node clustering, only noisy input node attributes are considered because there are no ground truth labels given for the clustering purposes.

### 5.1 Datasets

We evaluate BRGCL on five public benchmarks that are widely used for node representation learning, namely Cora, Citeseer, PubMed (Sen et al., 2008), Coauthor CS (Hamilton et al., 2017), ogbn-arxiv (Hu et al., 2020), Reddit (Hamilton et al., 2017), and AMiner-CS (Feng et al., 2020). Cora, Citeseer and PubMed are three most widely used citation networks. Coauthor CS is a co-authorship graph. The ogan-arxiv is a directed citation graph. The Reddit dataset is from Reddit posts whose node labels indicate the communities. AMiner-CS is extracted from the AMiner Citation Graph. We summarize the statistics of the datasets in Table 1. For all the experiments, we follow the default partition of training, validation, and test sets on each benchmark.

Table 1: The statistics of the datasets.

| Dataset | Nodes | Edges | Features | Classes |
|---------|-------|-------|----------|---------|
| **Cora** | 2,708 | 5,429 | 1,433 | 7 |
| **CiteSeer** | 3,327 | 4,732 | 3,703 | 6 |
| **PubMed** | 19,717 | 44,338 | 500 | 3 |
| **Coauthor CS** | 18,333 | 81,894 | 6,805 | 15 |
| **ogbn-arxiv** | 169,343 | 1,166,243 | 128 | 40 |
| **Reddit** | 232,965 | 11,606,919 | 602 | 41 |
| **AMiner-CS** | 593,486 | 6,217,004 | 100 | 18 |

### 5.2 Experimental Settings

Due to the fact that most public benchmark graph datasets do not come with corrupted labels or attribute noise, we manually inject noise into public datasets to evaluate our algorithm. We follow the commonly used label noise generation methods from the existing work (Han et al., 2020) to inject label noise. We generate noisy labels over all the classes according to a noise transition matrix $Q^{C \times C}$, where $Q_{ij}$ is the probability of nodes from class $i$ being flipped to class $j$ and $K$ is the ground truth class number. We consider two types of noise: (1) **symmetric**, where nodes from each class can be flipped to other classes with a uniform random probability such that $Q_{ij} = Q_{ji}$; (2) **asymmetric**, where mislabeling only occurs between similar

classes. The percentage of nodes with flipped labels is defined as the label noise level in our experiments. To evaluate the performance of our method with attribute noise, we randomly shuffle a certain percentage of input attributes for each node following (Ding et al., 2022). The percentage of the shuffled attributes is defined as the attribute noise level in our experiments.

**Compared Methods.** We compare BRGCL against semi-supervised node representation learning methods, including GCN (Kipf & Welling, 2017), GCE (Zhang & Sabuncu, 2018), $S^2$GC (Zhu & Koniusz, 2021), UnionNet (Li et al., 2021b), NRGNN (Dai et al., 2021), RTGNN (Qian et al., 2023), and GRAND+ (Feng et al., 2022b). We also compare BRGCL against the state-of-the-art GCL methods, including GraphCL (You et al., 2020), MVGRL (Hassani & Ahmadi, 2020), MERIT (Jin et al., 2021), and SUGRL (Mo et al., 2022). In addition, we compare BRGCL against three state-of-the-art robust contrastive learning methods, including ARIEL (Feng et al., 2022a), Jo-SRC (Yao et al., 2021), and Sel-CL (Li et al., 2022). Jo-SRC and Sel-CL were proposed for selecting clean samples for image data. Since their sample selection methods are general and not limited to the image domain, we adopt these two baselines in our experiments.

**ARIEL** (Feng et al., 2022a). ARIEL is a method proposed to improve the robustness of graph contrastive learning. Instead of adopting conventional graph data augmentation methods, ARIEL proposes to take the adversarial view generated by projected gradient descent attack as a new form of data augmentation. It also introduces an information regularization term to stabilize the training of InfoNCE loss for graph contrastive learning. In our experiments about ARIEL, we use the same GCN encoder as BRGCL to learn the node representations.

**Jo-SRC** (Yao et al., 2021). Jo-SRC is a robust contrastive learning method proposed for image classification. It selects clean samples for training by adopting the Jensen-Shannon divergence to measure the likelihood of each sample being clean. Because this method is a general selection strategy on the representation space, it can be adapted to selecting clean samples in the representation space of nodes in GCL. It also introduces a consistency regularization term to the contrastive loss to improve the robustness. To get a competitive and robust GCL baseline, we apply the sample selection strategy and the consistency regularization proposed by Jo-SRC to the state-of-the-art GCL methods MVGRL, MERIT, and SUGRL. We add the regularization term in Jo-SRC to the graph contrastive loss. The GCL encoders are trained only on the clean samples selected by Jo-SRC. We only report the best results for comparison, which are achieved by applying Jo-SRC to MERIT.

**Sel-CL** (Li et al., 2022). Sel-CL is a supervised contrastive learning proposed to learn robust pre-trained representations for image classification. It proposes to select confident contrastive pairs in the contrastive learning frameworks. Sel-CL first selects confident examples by measuring the agreement between learned representations and labels generated by label propagation with the cross-entropy loss. Next, Sel-CL selects contrastive pairs from those with selected confident examples in them. This method is also a general sample selection strategy on a learned representation space. As a result, we adapt Sel-CL to the node representation space to select confident pairs for GCL. In this process, they only select contrastive pairs whose representation similarity is higher than a dynamic threshold. In our experiments, we also adopt the confident contrastive pair selection strategy to the state-of-the-art GCL methods MVGRL, MERIT, and SUGRL. With the same GCL framework, GCL encoders are trained only on the confident pairs selected by Sel-CL. We only report the best results for comparison, which are achieved by applying Sel-CL to MERIT.

The training settings for different baselines are categorized into two setups, **unsupervised setup** and **supervised setup**. In the unsupervised setup, the training of the encoder does not use the ground truth label information. The node representations obtained by the encoder are then used for the downstream tasks, which are node classification and node clustering. In the supervised setup, the training of the encoder uses the ground truth label information. Our proposed BRGCL follows the unsupervised setup in all our experiments, and every baseline follows its corresponding setup by its nature. The implementation details about BRGCL are deferred to Section A of the appendix.

### 5.3 Evaluation Results

**Semi-supervised Node Classification with Label Noise.** We compare BRGCL against competing methods for semi-supervised node classification on input with two types of label noise. To show the robustness of BRGCL against label noise, we perform the experiments on graphs injected with different levels of label

Table 2: Performance comparison for node classification on PubMed, ogbn-arxiv, Cora, Citeseer, and Coauthor CS, with asymmetric label noise, symmetric label noise, and attribute noise. The baselines marked with * have their encoders trained with ground truth label information.

| Dataset | Methods | Noise Level 0 | 40 Asymmetric | 40 Symmetric | 40 Attribute | 60 Asymmetric | 60 Symmetric | 60 Attribute | 80 Asymmetric | 80 Symmetric | 80 Attribute |
|---|---|---|---|---|---|---|---|---|---|---|---|
| PubMed | GCN * | 0.790±0.007 | 0.584±0.022 | 0.574±0.012 | 0.595±0.012 | 0.405±0.025 | 0.386±0.011 | 0.488±0.013 | 0.305±0.022 | 0.295±0.013 | 0.423±0.013 |
| | S²GC * | 0.799±0.005 | 0.585±0.023 | 0.589±0.013 | 0.610±0.009 | 0.421±0.030 | 0.401±0.014 | 0.497±0.012 | 0.310±0.039 | 0.290±0.019 | 0.431±0.010 |
| | GCE | 0.792±0.009 | 0.589±0.018 | 0.581±0.011 | 0.590±0.014 | 0.430±0.012 | 0.399±0.012 | 0.491±0.010 | 0.311±0.021 | 0.301±0.011 | 0.424±0.012 |
| | UnionNET | 0.793±0.008 | 0.603±0.020 | 0.620±0.012 | 0.592±0.012 | 0.445±0.022 | 0.424±0.013 | 0.489±0.015 | 0.313±0.025 | 0.327±0.015 | 0.435±0.009 |
| | NRGNN | 0.797±0.008 | 0.602±0.022 | 0.618±0.013 | 0.603±0.008 | 0.443±0.012 | 0.434±0.012 | 0.499±0.009 | 0.330±0.023 | 0.325±0.013 | 0.433±0.011 |
| | SUGRL | 0.819±0.005 | 0.603±0.013 | 0.615±0.013 | 0.615±0.010 | 0.445±0.011 | 0.441±0.011 | 0.501±0.007 | 0.321±0.009 | 0.321±0.009 | 0.446±0.010 |
| | MVGRL | 0.794±0.003 | 0.599±0.012 | 0.613±0.012 | 0.606±0.008 | 0.441±0.013 | 0.433±0.013 | 0.496±0.010 | 0.322±0.012 | 0.312±0.012 | 0.438±0.010 |
| | MERIT | 0.801±0.004 | 0.593±0.011 | 0.612±0.011 | 0.613±0.011 | 0.447±0.012 | 0.443±0.012 | 0.497±0.009 | 0.328±0.011 | 0.323±0.011 | 0.445±0.009 |
| | Sel-Cl | 0.799±0.005 | 0.605±0.014 | 0.625±0.012 | 0.614±0.012 | 0.455±0.014 | 0.449±0.010 | 0.502±0.008 | 0.334±0.021 | 0.332±0.014 | 0.456±0.014 |
| | ARIEL | 0.800±0.003 | 0.610±0.013 | 0.622±0.010 | 0.615±0.011 | 0.453±0.012 | 0.453±0.012 | 0.502±0.014 | 0.331±0.014 | 0.336±0.018 | 0.457±0.013 |
| | Jo-SRC | 0.801±0.005 | 0.613±0.010 | 0.624±0.013 | 0.617±0.013 | 0.453±0.008 | 0.455±0.013 | 0.504±0.013 | 0.330±0.015 | 0.334±0.018 | 0.459±0.018 |
| | RTGNN | 0.797±0.004 | 0.610±0.008 | 0.622±0.010 | 0.614±0.012 | 0.455±0.010 | 0.455±0.011 | 0.501±0.011 | 0.335±0.013 | 0.338±0.017 | 0.452±0.013 |
| | GRAND+ | 0.837±0.006 | 0.610±0.011 | 0.624±0.013 | 0.617±0.013 | 0.453±0.008 | 0.453±0.011 | 0.503±0.010 | 0.331±0.014 | 0.337±0.013 | 0.458±0.014 |
| | BRGCL | 0.835±0.007 | **0.633±0.014** | **0.640±0.010** | **0.633±0.011** | **0.472±0.011** | **0.477±0.010** | **0.520±0.011** | **0.350±0.014** | **0.355±0.013** | **0.479±0.011** |
| ogbn-arxiv | GCN * | 0.717±0.003 | 0.401±0.014 | 0.421±0.014 | 0.478±0.010 | 0.336±0.011 | 0.346±0.021 | 0.339±0.012 | 0.286±0.022 | 0.256±0.010 | 0.294±0.013 |
| | S²GC * | 0.712±0.003 | 0.417±0.017 | 0.429±0.014 | 0.492±0.010 | 0.344±0.016 | 0.353±0.031 | 0.343±0.009 | 0.297±0.023 | 0.266±0.013 | 0.284±0.012 |
| | GCE | 0.720±0.004 | 0.410±0.018 | 0.428±0.008 | 0.480±0.014 | 0.348±0.019 | 0.344±0.019 | 0.342±0.015 | 0.310±0.014 | 0.260±0.011 | 0.275±0.015 |
| | UnionNET | 0.724±0.006 | 0.429±0.021 | 0.449±0.007 | 0.485±0.012 | 0.362±0.018 | 0.367±0.008 | 0.340±0.009 | 0.332±0.019 | 0.269±0.013 | 0.280±0.012 |
| | NRGNN | 0.721±0.006 | 0.449±0.014 | 0.466±0.009 | 0.485±0.012 | 0.371±0.020 | 0.379±0.008 | 0.342±0.011 | 0.330±0.018 | 0.271±0.018 | 0.300±0.010 |
| | SUGRL | 0.693±0.002 | 0.439±0.010 | 0.467±0.010 | 0.480±0.012 | 0.365±0.013 | 0.385±0.011 | 0.341±0.009 | 0.327±0.011 | 0.275±0.011 | 0.295±0.011 |
| | MVGRL | 0.713±0.002 | 0.443±0.009 | 0.461±0.009 | 0.481±0.008 | 0.372±0.012 | 0.382±0.012 | 0.339±0.009 | 0.329±0.013 | 0.274±0.013 | 0.290±0.012 |
| | MERIT | 0.717±0.004 | 0.442±0.009 | 0.463±0.009 | 0.483±0.010 | 0.368±0.011 | 0.381±0.011 | 0.341±0.012 | 0.324±0.012 | 0.272±0.010 | 0.304±0.009 |
| | Sel-Cl | 0.719±0.002 | 0.447±0.007 | 0.469±0.007 | 0.486±0.010 | 0.375±0.008 | 0.389±0.025 | 0.344±0.013 | 0.331±0.008 | 0.284±0.019 | 0.304±0.012 |
| | ARIEL | 0.717±0.004 | 0.448±0.013 | 0.471±0.013 | 0.482±0.011 | 0.379±0.014 | 0.384±0.015 | 0.342±0.015 | 0.334±0.014 | 0.280±0.013 | 0.300±0.010 |
| | Jo-SRC | 0.715±0.005 | 0.445±0.011 | 0.466±0.009 | 0.481±0.010 | 0.377±0.013 | 0.387±0.013 | 0.340±0.013 | 0.333±0.013 | 0.282±0.018 | 0.297±0.009 |
| | RTGNN | 0.718±0.004 | 0.443±0.012 | 0.464±0.012 | 0.484±0.014 | 0.380±0.011 | 0.384±0.013 | 0.340±0.017 | 0.335±0.011 | 0.285±0.015 | 0.301±0.006 |
| | GRAND+ | 0.725±0.004 | 0.445±0.008 | 0.466±0.011 | 0.481±0.011 | 0.378±0.010 | 0.385±0.012 | 0.344±0.010 | 0.332±0.010 | 0.282±0.016 | 0.303±0.009 |
| | BRGCL | 0.727±0.005 | **0.468±0.013** | **0.487±0.006** | **0.502±0.010** | **0.400±0.014** | **0.407±0.009** | **0.359±0.011** | **0.352±0.012** | **0.303±0.013** | **0.330±0.012** |
| Cora | GCN * | 0.817±0.005 | 0.547±0.015 | 0.636±0.007 | 0.639±0.008 | 0.405±0.014 | 0.517±0.010 | 0.439±0.012 | 0.265±0.012 | 0.354±0.014 | 0.317±0.013 |
| | S²GC * | 0.831±0.002 | 0.569±0.007 | 0.664±0.007 | 0.661±0.007 | 0.422±0.010 | 0.535±0.010 | 0.454±0.011 | 0.279±0.014 | 0.366±0.014 | 0.320±0.013 |
| | GCE | 0.819±0.004 | 0.573±0.011 | 0.652±0.008 | 0.650±0.014 | 0.449±0.011 | 0.509±0.011 | 0.445±0.015 | 0.280±0.013 | 0.353±0.013 | 0.325±0.015 |
| | UnionNET | 0.820±0.006 | 0.569±0.014 | 0.664±0.007 | 0.653±0.012 | 0.452±0.010 | 0.541±0.010 | 0.450±0.009 | 0.283±0.014 | 0.370±0.011 | 0.320±0.012 |
| | NRGNN | 0.822±0.006 | 0.571±0.019 | 0.676±0.007 | 0.645±0.012 | 0.470±0.014 | 0.548±0.014 | 0.451±0.011 | 0.282±0.022 | 0.373±0.012 | 0.326±0.010 |
| | SUGRL | 0.834±0.005 | 0.564±0.011 | 0.674±0.012 | 0.675±0.009 | 0.468±0.011 | 0.552±0.011 | 0.452±0.012 | 0.280±0.012 | 0.381±0.012 | 0.338±0.014 |
| | MVGRL | 0.829±0.007 | 0.566±0.009 | 0.672±0.009 | 0.655±0.011 | 0.455±0.014 | 0.545±0.014 | 0.445±0.012 | 0.275±0.014 | 0.379±0.014 | 0.330±0.014 |
| | MERIT | 0.831±0.005 | 0.560±0.008 | 0.670±0.008 | 0.671±0.009 | 0.467±0.013 | 0.547±0.013 | 0.450±0.014 | 0.277±0.013 | 0.385±0.013 | 0.335±0.009 |
| | Sel-Cl | 0.828±0.002 | 0.570±0.010 | 0.685±0.012 | 0.676±0.009 | 0.472±0.013 | 0.554±0.014 | 0.455±0.011 | 0.282±0.017 | 0.389±0.013 | 0.341±0.015 |
| | ARIEL | 0.829±0.004 | 0.573±0.013 | 0.681±0.010 | 0.675±0.009 | 0.471±0.012 | 0.553±0.012 | 0.455±0.014 | 0.284±0.014 | 0.389±0.013 | 0.343±0.013 |
| | Jo-SRC | 0.825±0.005 | 0.571±0.006 | 0.684±0.013 | 0.679±0.007 | 0.473±0.011 | 0.556±0.008 | 0.458±0.012 | 0.285±0.013 | 0.387±0.018 | 0.345±0.018 |
| | RTGNN | 0.828±0.003 | 0.570±0.010 | 0.682±0.008 | 0.678±0.011 | 0.474±0.011 | 0.555±0.010 | 0.457±0.009 | 0.280±0.011 | 0.386±0.014 | 0.342±0.016 |
| | GRAND+ | 0.853±0.006 | 0.570±0.009 | 0.682±0.007 | 0.678±0.011 | 0.472±0.010 | 0.554±0.008 | 0.456±0.012 | 0.284±0.015 | 0.387±0.015 | 0.345±0.013 |
| | BRGCL | 0.854±0.006 | **0.584±0.009** | **0.704±0.007** | **0.690±0.010** | **0.484±0.013** | **0.577±0.013** | **0.469±0.013** | **0.295±0.012** | **0.407±0.012** | **0.356±0.011** |
| Citeseer | GCN * | 0.703±0.005 | 0.475±0.023 | 0.501±0.013 | 0.529±0.009 | 0.351±0.014 | 0.341±0.014 | 0.372±0.011 | 0.291±0.022 | 0.281±0.019 | 0.290±0.014 |
| | S²GC * | 0.727±0.005 | 0.488±0.013 | 0.528±0.013 | 0.553±0.008 | 0.363±0.012 | 0.367±0.014 | 0.390±0.013 | 0.304±0.024 | 0.284±0.019 | 0.288±0.011 |
| | GCE | 0.705±0.004 | 0.490±0.016 | 0.512±0.014 | 0.540±0.014 | 0.362±0.015 | 0.352±0.010 | 0.381±0.009 | 0.309±0.012 | 0.285±0.014 | 0.285±0.011 |
| | UnionNET | 0.706±0.006 | 0.499±0.015 | 0.547±0.014 | 0.545±0.013 | 0.379±0.013 | 0.399±0.013 | 0.379±0.012 | 0.322±0.021 | 0.302±0.013 | 0.290±0.012 |
| | NRGNN | 0.710±0.006 | 0.498±0.015 | 0.546±0.015 | 0.538±0.011 | 0.382±0.016 | 0.412±0.016 | 0.377±0.012 | 0.336±0.021 | 0.309±0.018 | 0.284±0.009 |
| | SUGRL | 0.730±0.005 | 0.493±0.011 | 0.541±0.011 | 0.544±0.010 | 0.376±0.009 | 0.421±0.009 | 0.388±0.009 | 0.339±0.010 | 0.305±0.010 | 0.300±0.009 |
| | MVGRL | 0.726±0.007 | 0.491±0.013 | 0.541±0.013 | 0.540±0.008 | 0.379±0.013 | 0.420±0.013 | 0.386±0.011 | 0.341±0.016 | 0.301±0.016 | 0.282±0.011 |
| | MERIT | 0.740±0.007 | 0.496±0.012 | 0.536±0.012 | 0.542±0.010 | 0.383±0.011 | 0.425±0.011 | 0.387±0.008 | 0.344±0.014 | 0.301±0.014 | 0.295±0.009 |
| | Sel-Cl | 0.725±0.008 | 0.499±0.012 | 0.551±0.010 | 0.549±0.008 | 0.389±0.011 | 0.426±0.008 | 0.391±0.020 | 0.350±0.018 | 0.310±0.015 | 0.300±0.017 |
| | ARIEL | 0.729±0.007 | 0.500±0.008 | 0.550±0.013 | 0.548±0.008 | 0.391±0.009 | 0.427±0.012 | 0.389±0.014 | 0.349±0.014 | 0.307±0.013 | 0.299±0.013 |
| | Jo-SRC | 0.730±0.005 | 0.500±0.013 | 0.555±0.011 | 0.551±0.011 | 0.394±0.013 | 0.425±0.013 | 0.393±0.013 | 0.351±0.013 | 0.305±0.018 | 0.303±0.013 |
| | RTGNN | 0.746±0.008 | 0.498±0.007 | 0.556±0.007 | 0.550±0.012 | 0.392±0.010 | 0.424±0.013 | 0.390±0.014 | 0.348±0.017 | 0.308±0.016 | 0.302±0.011 |
| | GRAND+ | 0.746±0.004 | 0.497±0.010 | 0.553±0.010 | 0.552±0.011 | 0.390±0.013 | 0.422±0.013 | 0.387±0.013 | 0.348±0.013 | 0.309±0.014 | 0.302±0.012 |
| | BRGCL | 0.748±0.009 | **0.510±0.013** | **0.574±0.013** | **0.562±0.007** | **0.403±0.014** | **0.445±0.014** | **0.399±0.012** | **0.359±0.012** | **0.327±0.014** | **0.312±0.010** |
| Coauthor-CS | GCN * | 0.918±0.001 | 0.645±0.009 | 0.656±0.006 | 0.702±0.010 | 0.511±0.013 | 0.501±0.009 | 0.531±0.010 | 0.429±0.022 | 0.389±0.010 | 0.415±0.013 |
| | S²GC * | 0.918±0.001 | 0.657±0.012 | 0.663±0.006 | 0.713±0.010 | 0.516±0.013 | 0.514±0.009 | 0.556±0.009 | 0.437±0.020 | 0.396±0.010 | 0.422±0.012 |
| | GCE | 0.922±0.003 | 0.662±0.017 | 0.659±0.007 | 0.705±0.014 | 0.515±0.016 | 0.502±0.007 | 0.539±0.009 | 0.443±0.017 | 0.389±0.012 | 0.412±0.011 |
| | UnionNET | 0.918±0.002 | 0.669±0.023 | 0.671±0.013 | 0.706±0.012 | 0.525±0.011 | 0.529±0.011 | 0.540±0.012 | 0.458±0.015 | 0.401±0.011 | 0.420±0.007 |
| | NRGNN | 0.919±0.002 | 0.678±0.014 | 0.689±0.009 | 0.705±0.012 | 0.545±0.021 | 0.556±0.011 | 0.546±0.011 | 0.461±0.012 | 0.410±0.012 | 0.417±0.007 |
| | SUGRL | 0.922±0.005 | 0.675±0.010 | 0.695±0.010 | 0.714±0.006 | 0.550±0.011 | 0.560±0.011 | 0.561±0.007 | 0.449±0.011 | 0.411±0.011 | 0.429±0.008 |
| | MVGRL | 0.913±0.001 | 0.675±0.008 | 0.685±0.008 | 0.706±0.008 | 0.550±0.014 | 0.560±0.014 | 0.561±0.008 | 0.453±0.013 | 0.405±0.013 | 0.412±0.008 |
| | MERIT | 0.924±0.004 | 0.679±0.011 | 0.689±0.008 | 0.709±0.005 | 0.552±0.014 | 0.562±0.014 | 0.562±0.011 | 0.452±0.013 | 0.403±0.013 | 0.426±0.005 |
| | Sel-Cl | 0.922±0.008 | 0.684±0.009 | 0.694±0.012 | 0.714±0.010 | 0.557±0.013 | 0.568±0.013 | 0.566±0.010 | 0.457±0.013 | 0.412±0.017 | 0.425±0.009 |
| | ARIEL | 0.925±0.004 | 0.682±0.011 | 0.699±0.009 | 0.712±0.005 | 0.555±0.011 | 0.566±0.011 | 0.556±0.011 | 0.454±0.014 | 0.415±0.019 | 0.427±0.013 |
| | Jo-SRC | 0.921±0.005 | 0.684±0.011 | 0.695±0.004 | 0.709±0.007 | 0.560±0.011 | 0.566±0.011 | 0.561±0.009 | 0.456±0.013 | 0.410±0.018 | 0.428±0.010 |
| | RTGNN | 0.920±0.005 | 0.678±0.012 | 0.691±0.009 | 0.712±0.008 | 0.559±0.010 | 0.569±0.011 | 0.560±0.008 | 0.455±0.015 | 0.415±0.015 | 0.412±0.014 |
| | GRAND+ | 0.927±0.004 | 0.682±0.011 | 0.693±0.006 | 0.715±0.008 | 0.554±0.008 | 0.568±0.013 | 0.557±0.011 | 0.455±0.012 | 0.416±0.013 | 0.428±0.011 |
| | BRGCL | 0.929±0.006 | **0.694±0.013** | **0.718±0.008** | **0.733±0.009** | **0.570±0.014** | **0.587±0.011** | **0.585±0.012** | **0.465±0.012** | **0.434±0.015** | **0.444±0.012** |

noise ranging from 40% to 80% with a step of 20%. The classification follows the widely used semi-supervised setting (Kipf & Welling, 2017). It is noted that the labels are only used for the training of the classifier. The BRGCL encoder generates node representations, and the classifier for node classification is trained on these node representations.

In all the experiments, a two-layer MLP whose hidden dimension is 128 is used as the classifier. Detailed results on PubMed, ogbn-arxiv, Cora, Citeseer Coauthor CS are shown in Table 2, where we report the means of the accuracy of 10 runs and the standard deviation. Results on larger datasets, Reddit and AMiner-CS,

are shown in Table 6 in Section B of the appendix. It is observed from Table 2 and Table 6 that BRGCL outperforms all the baselines, including the methods using ground truth labels to train their encoders. By selecting confident nodes and computing robust prototypes using BEC, BRGCL outperforms all the baselines by an even larger margin with a larger label noise level. To verify the statistical significance of improvements, we show the $p$-values of t-test between BRGCL and the second best baseline. The $p$-values for all datasets with all noise levels for both symmetric label noise and asymmetric label noise are less than 0.05, suggesting the statistically significant improvement of BRGCL over baseline methods.

**Semi-supervised Node Classification with Attribute Noise.** We compare BRGCL with baselines for noisy input with attribute noise levels ranging from 40% to 80% with a step of 20%. The results on ogbn-arxiv are also illustrated in Figure 3 in Section B of the appendix. Detailed results on PubMed, ogbn-arxiv, Cora, Citeseer, and Coauthor CS are shown in Table 2, where we report the means of the accuracy of 10 runs and the standard deviation. The results clearly show that BRGCL is more robust to attribute noise compared to all the baselines for different noise levels. To verify the statistical significance of improvements, we show the $p$-values of t-test between BRGCL and the second best baseline. The $p$-values for all the datasets with all the levels of attribute noise are less than 0.05, suggesting the statistically significant improvement of BRGCL over the baseline methods.

**Node Clustering with Attribute Noise.** To further evaluate the robustness of node representation learned by BRGCL, we perform experiments on node clustering with attribute noise injected. We follow the same evaluation protocol as that in (Hassani & Ahmadi, 2020). K-means is applied on the learned node representations to obtain clustering results. We use accuracy (ACC) and normalized mutual information (NMI) as the performance metrics for clustering. The node clustering results for inputs with 60% attribute noise are shown in Table 7 in Section B of the appendix. We report the averaged clustering results and standard deviations over 20 times of execution. It is observed that node representation obtained by BRGCL is more robust to attribute noise for node clustering. To show the statistical significance of improvements, we also calculate $p$-values of the t-test between BRGCL and the second best baseline for each result. It is observed that BRGCL significantly improves the performance of node clustering as the $p$-values for ACC and NMI are less than 0.05 on all datasets.

## 5.4 Ablation Study on Confident Node Selection

To validate the effectiveness of confident node selection in BEC, we compare BRGCL with an ablation model which computes each prototype as the average of all the node representations assigned to that prototype cluster instead of the selected confident nodes. In order to explain the good performance of BRGCL under high noise levels, we perform the experiments of this ablation study with a noise level of 80 on Cora, Citeseer, and Pubmed. The results are shown in Table 3. It is observed that BRGCL outperforms the ablation model without node selection significantly, which demonstrates that the confident node selection largely mitigates the effects of noise in node classification.

Table 3: Ablation study on confident node selection in BRGCL with a noise level of 80.

| Datasets | Method | Noise | | |
|---|---|---|---|---|
| | | Asymmetric | Symmetric | Attribute |
| Cora | BRGCL w/o Node Selection | 0.277 | 0.385 | 0.335 |
| | BRGCL | **0.295** | **0.407** | **0.356** |
| Citeseer | BRGCL w/o Node Selection | 0.341 | 0.310 | 0.303 |
| | BRGCL | **0.359** | **0.327** | **0.312** |
| PubMed | BRGCL w/o Node Selection | 0.336 | 0.339 | 0.461 |
| | BRGCL | **0.350** | **0.355** | **0.479** |

## 5.5 Joint Training vs. Decoupled Training, and More Ablation Studies

We study the effectiveness of our decoupled training framework compared with jointly training the encoder and the classifier. We perform experiments on Cora, Citeseer, and Pubmed. The noise level is set to 80. It is observed from the results in Table 4 that decoupling the training of the classifier and the encoder is beneficial for mitigating the effects of label noise.

Moreover, we compare BRGCL with existing sample selection methods, including Co-teaching (Han et al., 2018) and Self-Training (Li et al., 2018), for node classification in Section C of the appendix. We further perform ablation study on the Bayesian nonparametric estimation of the number of prototypes in Section D of the appendix.

Table 4: Ablation study on joint training for node classification with label and attribute noise.

| Datasets | Method | Noise | | |
| --- | --- | --- | --- | --- |
| | | Asymmetric | Symmetric | Attribute |
| Cora | BRGCL w/o Node Selection | 0.289 | 0.400 | 0.351 |
| | BRGCL | **0.295** | **0.407** | **0.356** |
| Citeseer | BRGCL w/o Node Selection | 0.352 | 0.319 | 0.308 |
| | BRGCL | **0.359** | **0.327** | **0.312** |
| PubMed | BRGCL w/o Node Selection | 0.343 | 0.349 | 0.469 |
| | BRGCL | **0.350** | **0.355** | **0.479** |

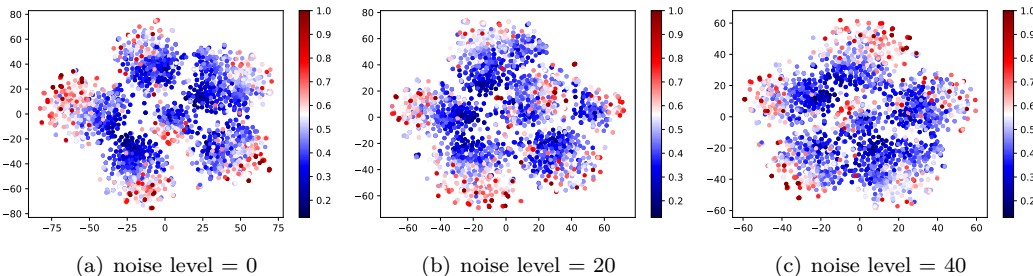

(a) noise level $= 0$  (b) noise level $= 20$  (c) noise level $= 40$

Figure 2: Visualization of confident nodes with different levels of attribute noise for semi-supervised node classification on Citeseer.

### 5.6 Visualization of Confidence Score

We visualize the confident nodes selected by BEC in the embedding space of the learned node representations in Figure 2. The node representations are visualized by the t-SNE figure. Each mark in t-SNE represents the representation of a node, and the color of the mark denotes the normalized confidence score of that node. Given the confidence scores $\{\alpha_i\}_{i=1}^N$ of all the nodes, let $\alpha_{\min}$ and $\alpha_{\max}$ be the minimum and the maximum confidence scores, then the normalized confidence score for a node $v_i$ is $\bar{\alpha}_i = (\alpha_i - \alpha_{\min})/(\alpha_{\max} - \alpha_{\min})$ for $i \in [N]$. The visualization results are shown for different levels of attribute noise. It can be observed from Figure 2 that confident nodes, which are redder in Figure 2, are well separated in the embedding space. With a higher level of attribute noise, the bluer nodes from different clusters blended around the cluster boundaries. In contrast, the redder nodes are still well separated and far away from the cluster boundaries, which leads to more robustness and better performance in downstream tasks.

## 6  Conclusions

In this paper, we propose a novel node representation learning method termed Bayesian Robust Graph Contrastive Learning (BRGCL) which improves the robustness of node representations by a novel Bayesian nonparametric algorithm, Bayesian nonparametric Estimation of Confidence (BEC). We evaluate the performance of BRGCL with comparison to competing baselines on semi-supervised node classification and node clustering, where graph data are corrupted with noise in either the labels or the node attributes. Experimental results demonstrate that BRGCL generates more robust node representations with better performance than the current state-of-the-art node representation learning methods.

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

## A  Implementation Details

We implement our proposed framework in PyTorch. All the experiments are conducted on a NVIDIA Tesla A100 GPU. The hidden dimension $m$ of the BRGCL encoder $g$ is fixed to 512. In the node-wise contrastive loss, we set the number of negative samples to 512 to avoid the out-of-memory issue in large datasets. We perform a grid search for the learning rate in $\{1 \times 10^{-5}, 5 \times 10^{-5}, 1 \times 10^{-4}, 5 \times 10^{-4}, 1 \times 10^{-3}, 5 \times 10^{-3}, 1 \times 10^{-2}, 5 \times 10^{-2}, 1 \times 10^{-1}, 5 \times 10^{-1}\}$ on different datasets with different types of noise. The weight decay is set to $5 \times 10^{-6}$, and $t_{\max}$ is set to 500. Adam optimizer is used in our training.

**Tuning Hyper-Parameters by Cross-Validation.** To find the optimal values of the hyper-parameters $\xi$ in (5) and $\gamma_0$ in (8), we perform cross-validations on 20% of the training data to decide the value of $\xi$ and $\gamma_0$. The value of $\xi$ is selected from $\{0.1, 0.15, 0.2, 0.25, 0.3, 0.35, 0.4, 0.45, 0, 5\}$. The value of $\gamma_0$ is selected from $\{0.1, 0.2, 0.3, 0.4, 0.5, 0.6, 0.7, 0.8, 0.9\}$. The selected values for $\xi$ and $\gamma_0$ on each dataset are shown in Table 5.

Table 5: Selected hyper-parameters for each dataset.

| Dataset | Cora | Citeseer | PubMed | Coauthor CS | ogbn-arxiv |
|---|---|---|---|---|---|
| $\xi$ | 0.20 | 0.15 | 0.35 | 0.40 | 0.25 |
| $\gamma_0$ | 0.3 | 0.5 | 0.7 | 0.4 | 0.4 |

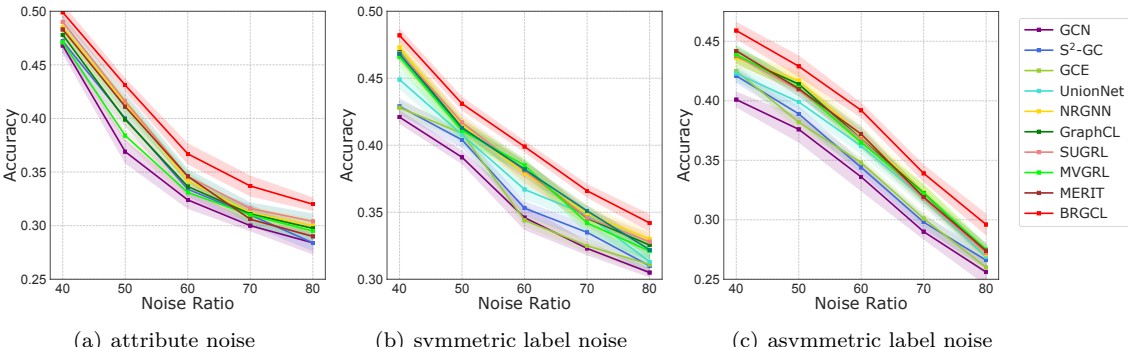

(a) attribute noise      (b) symmetric label noise      (c) asymmetric label noise

Figure 3: Performance comparisons on semi-supervised node classification on ogbn-arxiv with different levels of attribute noise, symmetric label noise, and asymmetric label noise. The shaded areas around the lines denote the standard deviation of the classification accuracy.

# B More Results about Node Classification and Clustering

We provide additional experimental results for node classification and node clustering in this section. The results of different methods with respect to different levels of symmetric and asymmetric label noise and attribute noise on the ogbn-arxiv dataset for node classification are illustrated in Figure 3. The results about semi-supervised node classification with label noise and attribute noise on two large datasets, Reddit and AMiner-CS, are shown in Table 6. Moreover, Table 7 shows the results about node clustering with 60% attribute noise.

Table 6: Performance comparison for node classification with asymmetric label noise, symmetric label noise, and attribute noise on large scale dataset. The baselines marked with * have their encoders trained with ground truth label information.

| Dataset | Methods | Noise Level | | | | | | | | | |
| | | 0 | 40 | | | 60 | | | 80 | | |
| | | - | Asymmetric | Symmetric | Attribute | Asymmetric | Symmetric | Attribute | Asymmetric | Symmetric | Attribute |
|---|---|---|---|---|---|---|---|---|---|---|---|
| Reddit | GCN * | 0.917±0.004 | 0.601±0.015 | 0.621±0.012 | 0.678±0.011 | 0.536±0.010 | 0.546±0.019 | 0.539±0.014 | 0.486±0.024 | 0.456±0.009 | 0.494±0.014 |
| | S²GC * | 0.912±0.002 | 0.617±0.018 | 0.629±0.016 | 0.692±0.008 | 0.544±0.015 | 0.553±0.030 | 0.543±0.010 | 0.497±0.022 | 0.466±0.012 | 0.484±0.011 |
| | GCE | 0.920±0.005 | 0.610±0.019 | 0.628±0.007 | 0.680±0.013 | 0.548±0.020 | 0.544±0.020 | 0.542±0.014 | 0.510±0.013 | 0.460±0.012 | 0.475±0.014 |
| | UnionNET | 0.924±0.007 | 0.629±0.022 | 0.649±0.006 | 0.685±0.011 | 0.562±0.019 | 0.567±0.009 | 0.540±0.008 | 0.532±0.018 | 0.469±0.014 | 0.480±0.013 |
| | NRGNN | 0.936±0.007 | 0.664±0.015 | 0.681±0.011 | 0.700±0.013 | 0.586±0.019 | 0.594±0.009 | 0.557±0.012 | 0.545±0.017 | 0.486±0.020 | 0.515±0.010 |
| | SUGRL | 0.908±0.003 | 0.654±0.009 | 0.682±0.011 | 0.695±0.012 | 0.580±0.014 | 0.600±0.010 | 0.556±0.008 | 0.542±0.010 | 0.490±0.010 | 0.510±0.011 |
| | MVGRL | 0.928±0.007 | 0.658±0.010 | 0.676±0.008 | 0.696±0.009 | 0.587±0.013 | 0.597±0.011 | 0.554±0.008 | 0.544±0.014 | 0.489±0.015 | 0.505±0.011 |
| | MERIT | 0.932±0.003 | 0.657±0.010 | 0.678±0.009 | 0.698±0.009 | 0.583±0.010 | 0.596±0.010 | 0.556±0.011 | 0.539±0.012 | 0.487±0.009 | 0.519±0.008 |
| | Sel-Cl | 0.931±0.003 | 0.659±0.008 | 0.681±0.007 | 0.698±0.009 | 0.587±0.009 | 0.601±0.026 | 0.556±0.014 | 0.543±0.009 | 0.496±0.018 | 0.516±0.013 |
| | ARIEL | 0.929±0.003 | 0.660±0.014 | 0.683±0.012 | 0.694±0.010 | 0.591±0.015 | 0.596±0.016 | 0.554±0.016 | 0.546±0.015 | 0.492±0.014 | 0.512±0.009 |
| | Jo-SRC | 0.927±0.006 | 0.657±0.010 | 0.678±0.008 | 0.693±0.011 | 0.589±0.014 | 0.599±0.012 | 0.552±0.014 | 0.545±0.014 | 0.494±0.017 | 0.509±0.008 |
| | RTGNN | 0.930±0.003 | 0.655±0.013 | 0.676±0.013 | 0.696±0.015 | 0.592±0.010 | 0.596±0.014 | 0.552±0.018 | 0.547±0.012 | 0.497±0.016 | 0.513±0.007 |
| | GRAND+ | 0.937±0.005 | 0.657±0.007 | 0.678±0.012 | 0.693±0.012 | 0.590±0.009 | 0.597±0.013 | 0.556±0.012 | 0.544±0.011 | 0.494±0.017 | 0.515±0.010 |
| | BRGCL | **0.939±0.004** | **0.680±0.012** | **0.699±0.007** | **0.714±0.009** | **0.612±0.015** | **0.619±0.010** | **0.571±0.010** | **0.564±0.013** | **0.515±0.012** | **0.542±0.013** |
| AMiner-CS | GCN * | 0.492±0.004 | 0.385±0.023 | 0.386±0.013 | 0.419±0.009 | 0.307±0.014 | 0.337±0.014 | 0.327±0.011 | 0.259±0.014 | 0.272±0.013 | 0.288±0.023 |
| | S²GC * | 0.502±0.002 | 0.396±0.013 | 0.418±0.013 | 0.443±0.008 | 0.320±0.012 | 0.346±0.014 | 0.342±0.013 | 0.270±0.012 | 0.281±0.012 | 0.293±0.013 |
| | GCE | 0.513±0.005 | 0.408±0.016 | 0.401±0.014 | 0.430±0.014 | 0.318±0.015 | 0.338±0.010 | 0.336±0.009 | 0.266±0.015 | 0.276±0.012 | 0.290±0.016 |
| | UnionNET | 0.517±0.007 | 0.416±0.015 | 0.435±0.014 | 0.435±0.013 | 0.335±0.013 | 0.348±0.013 | 0.334±0.012 | 0.265±0.014 | 0.285±0.015 | 0.295±0.015 |
| | NRGNN | 0.516±0.007 | 0.407±0.015 | 0.435±0.015 | 0.428±0.011 | 0.339±0.016 | 0.361±0.016 | 0.333±0.012 | 0.274±0.016 | 0.299±0.012 | 0.302±0.015 |
| | SUGRL | 0.523±0.003 | 0.413±0.011 | 0.430±0.011 | 0.433±0.010 | 0.333±0.009 | 0.371±0.009 | 0.343±0.009 | 0.272±0.009 | 0.294±0.013 | 0.300±0.011 |
| | MVGRL | 0.525±0.007 | 0.411±0.013 | 0.430±0.013 | 0.429±0.008 | 0.336±0.013 | 0.369±0.013 | 0.341±0.011 | 0.285±0.013 | 0.304±0.013 | 0.311±0.013 |
| | MERIT | 0.525±0.003 | 0.415±0.012 | 0.425±0.012 | 0.431±0.010 | 0.340±0.011 | 0.373±0.011 | 0.342±0.008 | 0.290±0.011 | 0.310±0.013 | 0.305±0.012 |
| | Sel-Cl | 0.520±0.003 | 0.408±0.012 | 0.440±0.010 | 0.438±0.008 | 0.346±0.011 | 0.376±0.008 | 0.346±0.020 | 0.292±0.011 | 0.309±0.014 | 0.303±0.012 |
| | ARIEL | 0.513±0.003 | 0.419±0.008 | 0.439±0.013 | 0.437±0.008 | 0.348±0.009 | 0.377±0.012 | 0.345±0.014 | 0.294±0.009 | 0.316±0.011 | 0.301±0.008 |
| | Jo-SRC | 0.511±0.006 | 0.419±0.013 | 0.444±0.011 | 0.440±0.011 | 0.351±0.013 | 0.375±0.013 | 0.349±0.013 | 0.290±0.013 | 0.315±0.015 | 0.300±0.014 |
| | RTGNN | 0.516±0.003 | 0.417±0.007 | 0.445±0.007 | 0.439±0.012 | 0.349±0.010 | 0.373±0.013 | 0.346±0.014 | 0.288±0.010 | 0.320±0.014 | 0.311±0.021 |
| | GRAND+ | **0.540±0.005** | 0.413±0.010 | 0.438±0.010 | 0.442±0.011 | 0.347±0.013 | 0.371±0.013 | 0.343±0.013 | 0.280±0.010 | 0.313±0.015 | 0.305±0.013 |
| | BRGCL | 0.533±0.004 | **0.428±0.012** | **0.457±0.007** | **0.455±0.009** | **0.362±0.015** | **0.389±0.010** | **0.360±0.010** | **0.309±0.011** | **0.336±0.014** | **0.322±0.015** |

# C Comparisons to Existing Sample Selection Methods

In this subsection, we compare BRGCL against existing sample selection methods, including Co-teaching (Han et al., 2018) and Self-Training (Li et al., 2018) for node classification with symmetric label noise. Co-teaching maintains two networks to select clean samples for each other. Self-Training finds nodes with the most confident pseudo labels, and it augments the labeled training data by incorporating confident nodes with

Table 7: Node clustering performance comparison on benchmark datasets with 60% input attribute noise. The *p*-values of the t-test between BRGCL and the second best baseline are listed in the last row of the table.

| Methods | Cora | | Citeseer | | PubMed | | Coauthor CS | | ogbn-arxiv | |
|---|---|---|---|---|---|---|---|---|---|---|
| | ACC | NMI | ACC | NMI | ACC | NMI | ACC | NMI | ACC | NMI |
| | | | | | Supervised | | | | | |
| GCN | 57.4±0.61 | 44.7±0.57 | 57.1±0.65 | 35.4±0.34 | 56.9±0.99 | 28.3±0.46 | 52.8±0.74 | 52.6±0.95 | 43.7±1.19 | 49.6±0.84 |
| S²GC | 58.4±0.72 | 47.3±0.79 | 58.3±0.82 | 35.1±0.65 | 57.4±0.89 | 28.3±0.31 | 53.6±0.99 | 54.0±1.09 | 45.2±0.97 | 50.2±0.70 |
| NRGNN | 61.1±0.73 | 47.8±0.93 | 57.8±0.77 | 36.2±0.71 | 57.1±1.03 | 29.1±0.59 | 53.3±0.87 | 54.1±1.02 | 44.1±1.04 | 50.1±0.85 |
| | | | | | Unsupervised | | | | | |
| K-means | 39.9±0.94 | 26.9±0.88 | 44.8±0.59 | 26.8±1.76 | 49.0±1.45 | 29.3±1.49 | 25.4±1.76 | 14.6±1.86 | 24.3±1.76 | 27.9±1.86 |
| GAE | 49.1±0.95 | 36.9±0.67 | 33.2±0.64 | 16.4±1.36 | 56.6±0.87 | 26.1±0.65 | 39.6±1.25 | 38.9±1.40 | 34.5±1.14 | 36.4±1.32 |
| ARVGA | 53.8±1.01 | 39.0±0.59 | 45.2±0.82 | 24.2±0.78 | 57.2±0.69 | 27.0±0.46 | 49.8±0.65 | 48.3±1.13 | 40.2±0.77 | 44.3±1.03 |
| GALA | 63.3±0.78 | 50.0±0.68 | 59.4±0.80 | 35.8±0.88 | 57.1±0.79 | 29.1±0.17 | 52.5±1.03 | 53.8±0.98 | 45.2±0.97 | 50.5±0.79 |
| GraphCL | 61.2±0.96 | 49.1±0.79 | 58.3±0.88 | 34.9±1.02 | 57.3±0.89 | 29.1±0.49 | 53.2±0.88 | 54.2±1.14 | 43.9±0.97 | 49.3±1.03 |
| MVGRL | 62.5±0.79 | 50.5±0.63 | 59.2±0.79 | 35.7±0.76 | 57.6±0.70 | 29.6±0.55 | 54.1±0.87 | 55.2±1.02 | 45.1±0.89 | 50.2±0.95 |
| MERIT | 63.0±0.87 | 51.1±0.75 | 59.2±0.69 | 36.1±0.45 | 57.9±0.80 | 30.2±0.42 | 54.8±0.87 | 56.4±0.79 | 45.4±0.78 | 51.0±0.81 |
| BRGCL | **63.8±0.69** | **51.9±0.81** | **60.3±0.79** | **37.1±0.63** | **58.8±0.59** | **30.9±0.85** | **56.1±0.64** | **58.2±0.96** | **46.5±0.86** | **52.2±0.91** |
| *p*-value | 0.0014 | 0.0021 | 0.0231 | 0.0030 | 0.0401 | 0.0154 | 0.0075 | 0.0102 | 0.0112 | 0.0144 |

their pseudo labels into the existing training data. The results are shown in Table 8. It can be observed that BRGCL greatly outperforms these two competing sample selection methods.

Table 8: Performance comparison against Co-teaching (Han et al., 2018) and Self-training (Li et al., 2018) on node classification with different levels of symmetric label noise.

| Dataset | Methods | Noise Level | | | | |
|---|---|---|---|---|---|---|
| | | 40 | 50 | 60 | 70 | 80 |
| Cora | Self-training | 0.664±0.012 | 0.584±0.007 | 0.532±0.013 | 0.459±0.011 | 0.368±0.012 |
| | Co-teaching | 0.668±0.011 | 0.593±0.011 | 0.527±0.010 | 0.465±0.010 | 0.367±0.017 |
| | BRGCL | **0.704±0.007** | **0.622±0.009** | **0.577±0.013** | **0.500±0.014** | **0.407±0.012** |
| Citeseer | Self-training | 0.541±0.014 | 0.465±0.013 | 0.397±0.013 | 0.347±0.016 | 0.301±0.022 |
| | Co-teaching | 0.522±0.018 | 0.461±0.011 | 0.383±0.011 | 0.338±0.014 | 0.299±0.020 |
| | BRGCL | **0.574±0.013** | **0.496±0.011** | **0.445±0.014** | **0.368±0.013** | **0.327±0.014** |
| PubMed | Self-training | 0.597±0.019 | 0.507±0.011 | 0.419±0.021 | 0.380±0.020 | 0.345±0.023 |
| | Co-teaching | 0.584±0.013 | 0.499±0.015 | 0.403±0.014 | 0.371±0.011 | 0.342±0.022 |
| | BRGCL | **0.640±0.010** | **0.530±0.010** | **0.477±0.010** | **0.399±0.012** | **0.355±0.013** |
| Coauthor CS | Self-training | 0.672±0.010 | 0.614±0.012 | 0.542±0.013 | 0.462±0.015 | 0.397±0.015 |
| | Co-teaching | 0.666±0.012 | 0.610±0.011 | 0.529±0.015 | 0.451±0.013 | 0.404±0.019 |
| | BRGCL | **0.718±0.008** | **0.638±0.009** | **0.587±0.011** | **0.480±0.011** | **0.434±0.015** |
| ogbn-arxiv | Self-training | 0.462±0.012 | 0.413±0.014 | 0.368±0.018 | 0.328±0.014 | 0.276±0.020 |
| | Co-teaching | 0.437±0.024 | 0.406±0.011 | 0.359±0.016 | 0.322±0.012 | 0.282±0.025 |
| | BRGCL | **0.487±0.006** | **0.432±0.009** | **0.407±0.009** | **0.344±0.012** | **0.303±0.013** |

Table 9: Ablation study on the Bayesian nonparametric estimation of the number of prototypes.

| Datasets | #Prototypes | Noise | | |
|---|---|---|---|---|
| | | Asymmetric | Symmetric | Attribute |
| Cora | 5 | 0.284 | 0.390 | 0.349 |
| | 7 (Ground Truth Class Number) | 0.288 | 0.399 | 0.350 |
| | 10 | 0.289 | 0.403 | 0.352 |
| | 13 (Estimated by BRGCL) | **0.295** | **0.407** | **0.356** |
| | 15 | 0.293 | 0.402 | 0.354 |
| Citeseer | 5 | 0.354 | 0.325 | 0.309 |
| | 6 (Ground Truth Class Number) | 0.355 | 0.324 | 0.310 |
| | 9 (Estimated by BRGCL) | **0.359** | **0.327** | **0.312** |
| | 10 | 0.358 | **0.327** | 0.310 |
| | 15 | 0.353 | 0.321 | 0.308 |
| PubMed | 3 (Ground Truth Class Number) | 0.342 | 0.349 | 0.470 |
| | 5 | 0.343 | 0.352 | 0.471 |
| | 10 | 0.348 | 0.354 | 0.476 |
| | 11 (Estimated by BRGCL) | **0.350** | **0.355** | **0.479** |
| | 15 | 0.349 | 0.354 | 0.478 |

# D  Ablation Study on the Bayesian Nonparametric Estimation of the Number of Prototypes

To validate the effectiveness of using the Bayesian nonparametric method to infer the number of cluster prototypes in BEC, we compare BRGCL with ablation models that manually set the number of cluster prototypes. We perform the experiments on Cora, Citeseer, and Pubmed with a noise level of 80. Herein, we compare the Bayesian nonparametric method with methods that manually set the number of prototypes to 5, 10, and 15, as well as the ground truth number of classes in each dataset. It is observed from Table 9 that estimating the number of prototypes by the Bayesian nonparametric method in BEC usually achieves the best performance on each dataset. The results show that the Bayesian nonparametric method can find a suitable number of prototypes for robustness to noise.

