# OpenReview forum: "Bayesian Robust Graph Contrastive Learning"
_TMLR — Withdrawn by Authors_

### Review · Reviewer_feP5 · 2024-04-19

**Summary Of Contributions:**

The paper is about a 2-step unsupervised method for node prediction on graph. The idea is to find a set of trustworthy labels to be used as a noise-robust training set. Label estimates are considered reliable if they are closed to a cluster's centroid.

**Audience:**

Yes

**Broader Impact Concerns:**

I do not think this work may raise any ethical concern.

**Claims And Evidence:**

Yes

**Requested Changes:**

Questions:
- In the abstract, why does unsupervised learning imply noise robustness?
- What does “such confident nodes are far away from the class/cluster boundaries” mean? What are the conditions for the assumption to hold?
- Some part of the pseudo labels inference is based on clustering. Which algorithm is used to obtain the clusters?
- Could you add the reason the proposed method is called, “contrastive”, in the introduction?
- How does the performance depend on the techniques used to generate the contrastive views of Section 4.1?
- What is the motivation for estimating the label confidence by comparing the output of BPL and “label-propagation”?
- BRGCL is an unsupervised learning algorithm. Why are all experiments of Section 5.3 in the semi-supervised setup?

**Strengths And Weaknesses:**

Strengths:
- I like the idea of finding a set of trustworthy data points and estimating their confidence. The approach may be generalized to practical scenarios beyond graph machine learning.
- Unsupervised learning on graphs has many real-world applications.
- In the provided empirical evaluation, the proposed method largely outperforms existing node prediction algorithms.

Weaknesses:
- The presentation may be improved. As the approach combines different steps, it would be good to have a more structured presentation. For example, it would be nice to see an end-to-end summary of the scheme.
- The assumptions that about the existence and location of trustworthy nodes may be justified better, e.g. with a practical example.
- BPL and BEC focus on label estimation. The authors should explain how the algorithm is expected to handle noise in the attributes.
- The differences compared to existing work should be clarified. I appreciate the detailed review of Section 2.2 but would have expected some explanation on the difference from existing Graph Contrastive Learning approaches, e.g. MERIT, SUGRL, and RS-GNN.
- The contribution looks like an (empirically efficient) pipeline of different existing techniques rather than a new principled methodology. The authors may clarify what is new in each step of their scheme.

---

### Review · Reviewer_ojCU · 2024-04-22

**Summary Of Contributions:**

The paper proposes the Bayesian Robust Graph Contrastive Learning (BRGCL) aiming to enhance robustness of node representations for GNNs. Specifically, the procedure trains an encoder on noisy graph data and exploits a non-parametric Dirichlet Process mixture model to group nodes and assign confidence based on the extracted node-specific responsibilities. The approach explores contrastive learning using the extracted centroids based on the established confident nodes to guide the encoder to be robust to label and attribute noise. The approach is on real graphs found to provide superior performance to conventional GNN training in node classification and clustering.

The proposed approach is rather convoluted and requires the following steps:

1)	Train a conventional encoder using contrastive learning guided by consistency of two views by use of infoNCE contrastive loss L_node and cluser constrative loss L_propotype.

2)	Learn a non-parametric clustering procedure using the zero-variance limit of a Dirichlet mixture model to identify clusters – use cross-validation to quantify when new clusters are formed based on distances to existing centroids.

3)	Use the graph to perform cluster label propagation and quantify confident nodes based on this label propagation.

4)	Guide the encoder using the prototype contrastive loss using updated centroids defined based on an annealed threshold level of the confident nodes.

5)	Keep training using 2-4 with the annealing procedure on what is deemed confident nodes until a sufficient number of training steps.
6)	Train supervised an MLP based on the encoded space for node classification.

Step 2 and 4 are further tuned using cross-validation on an external validation set to guide when new clusters are formed as well as when nodes are deemed confident (as such, step 6 is also used for tuning these hyper-parameters).

**Audience:**

Yes

**Broader Impact Concerns:**

Broader impact is not discussed, however, there are arguably also no direct concerns regarding the executed research. Potentially aspects to discuss in terms of problematic uses of graph representation learning for profiling and influencing as well as surveillance purposes could be discussed, but I do not see the scope of this research necessitating this.

**Claims And Evidence:**

Yes

**Requested Changes:**

How is cross-validation invoked to choose the value of ξ in the context of node clustering?

The L_prototype relies on a parameter \tau defining the scale of similarity – how was this parameter tuned?
What prevents the clustering procedure to form clusters of outliers, i.e small distinct clusters of noisy nodes? This would deteriorate performance and also violate the overall hypothesis that nodes with high confidence for a cluster is noise robust.

In the experiments the following is stated “Due to the fact that most public benchmark graph datasets do not come with corrupted labels or attribute noise, we manually inject noise into public datasets to evaluate our algorithm”. This seems a rather synthetic experimental setting and it would strengthen the paper to consider datasets in which the procedure works better without this need to inject artificial noise – is this possible to show?

In the experimentation it is stated that “Our proposed BRGCL follows the unsupervised setup in all our
experiments, and every baseline follows its corresponding setup by its nature. “ However, as cross-validation is used to tune hyper-parameters I assume this step will be supervised and thus the method not entirely unsupervised – this needs to be clarified.


There are no error bars in the results of Table 3 and Table 4 whereas the standard deviation reported in Table 2 is unclear what is across – different training/test splits or based on different random initializations for the same fixed training/test split? Please clarify.

What is the performance metric used for node classification? – this is not explained. How does this performance compare to simple shallow embedding procedures and contentional graph representation learning modeling approaches – see also
Nakis, Nikolaos, et al. "A Hierarchical Block Distance Model for Ultra Low-Dimensional Graph Representations." IEEE Transactions on Knowledge and Data Engineering (2023).

**Strengths And Weaknesses:**

Strengths

•	The approach appears to work well in practice and the results are compelling.

•	The paper has a fine literature review of related approaches and how the paper well positioned herein.

•	The Visual illustrations help understand the somewhat convoluted modeling approach.

Weaknesses

•	The approach appears rather heuristic and contains many steps that are combined (see also algorithm 1) which makes the approach not very principled and somewhat complicated to use in practice (see summary above outlining the proposed procedure).

•	It is unclear that the proposed procedure is fully unsupervised as it includes cross-validation supervised steps for hyperparameter tuning.

•	The motivation why clusters cannot be driven by outliers is unclear.

---

### Review · Reviewer_TSE7 · 2024-04-27

**Summary Of Contributions:**

This paper proposed a novel and robust method, Bayesian Robust Graph Contrastive Learning (BRGCL), for training a GNN encoder to learn robust node representations. It introduced an unsupervised encoder that estimates confident nodes and computes robust cluster prototypes of node representations using a Bayesian nonparametric method, then it implemented prototypical contrastive learning between node representations and robust cluster prototypes to enhance performance and robustness of learned node representations.

**Audience:**

No

**Claims And Evidence:**

No

**Requested Changes:**

Solve my previous concerns

**Strengths And Weaknesses:**

Weakness:

(1) The paper only considers GCN, and it is unknown how to use their method for more advanced structures such as Graph Transformers.
(2) From Table, I can easily see that the improvement compare to other baselines is very limited. Moreover, giving the s.t.d it is hard to claim their method is better than baselines.
(3) The key idea, Bayesian non-parametric prototype learning, is incremental and trivial. I cannot see any challenge or the difference with previous study (non for GNN). It seems like the method here is independent on the graph structure. I want to see a better idea which come from the graph structure, rather than a direct combination with GCN.

---

### Note · Authors · 2024-04-27

I have read and agree with the venue's withdrawal policy on behalf of myself and my co-authors.